# Computational Modeling and Simulation to Increase Laser Shooting Accuracy of Autonomous LEO Trackers

**Jose M. Gambi** [1,*] , **Maria L. Garcia del Pino** [2] , **Jonathan Mosser** [3] **and Ewa B. Weinmüller** [3]

1    Gregorio Millan Institute, University Carlos III de Madrid, 28911 Leganes, Spain
2    Department of Mathematics, IES Alpajes, 28300 Aranjuez, Spain; lgarciadelpino@educa.madrid.org
3    Institute for Analysis and Scientific Computing, TU Wien, Vienna 1040, Austria;
     jonathan.mosser@asc.tuwien.ac.at (J.M.); ewa.weinmueller@asc.tuwien.ac.at (E.B.W.)
*    Correspondence: gambi@math.uc3m.es

**Abstract:** In this paper, we introduce a computational procedure that enables autonomous LEO laser trackers endowed with INSs to increase the current accuracy when shooting at middle distant medium-size LEO debris targets. The code is designed for the trackers to throw the targets into the atmosphere by means of ablations. In case that the targets are eclipsed to the trackers by the Earth, the motions of the trackers and targets are modeled by equations that contain post-Newtonian terms accounting for the curvature of space. Otherwise, when the approaching targets become visible for the trackers, we additionally use more accurate equations, which allow to account for the local bending of the laser beams aimed at the targets. We observe that under certain circumstances the correct shooting configurations that allow to safely and efficiently shoot down the targets, differ from the current estimations by distances that may be larger than the size of many targets. In short, this procedure enables to estimate the optimal shooting instants for any middle distant medium-size LEO debris target.

**Keywords:** autonomous LEO trackers; debris targets; laser ablation; Earth post-Newtonian framework





## 1. Introduction

It is well known that there is a very large number (thousands) of LEO debris objects, and that one of the most promising proposals to get rid of those of middle size is to throw them into the atmosphere by means of laser ablations (see e.g., [1]). Hence, one of the most challenging issues from the orbital standpoint is to guarantee the appropriate accuracy for the laser beams that are shot from space-based LEO trackers.

The aim of this work is to introduce a procedure that allows to substantially decrease the shooting errors of the trackers. The code is especially designed for any standard autonomous tracker endowed with atomic clocks and inertial navigation systems (INSs) to determine the optimal shooting instants that allow to confidently shoot down the targets. In fact, the code provides not only these instants, but also the pointing directions for the trackers to safely and efficiently reach the approaching targets in order to throw them into the atmosphere by means of accurate laser ablations.

To this end, the procedure contains Earth Centered Inertial (ECI) and relative Newtonian (N) orbital equations for a spherical Earth and post-Newtonian (p-N) ECI equations for the Earth Schwarzschild field, along with two families of relative equations, which are matched to be compatible with the ECI p-N equations and with the tracking equations in [2]. Consequently, the code provides non-standard corrections both for the optimal instants and locations to shoot down the targets.

In the cases considered in this paper it turns out that the p-N pointing deviations of the preliminary N target's locations are rather large, in particular for small targets. The difference between the N and p-N locations may be in general larger than their size. But even more important are the corrections to the instants for shooting down the targets.

They are substantial and measurable by means of the atomic clocks on board of trackers. Therefore, these corrections can be complementary to the corrections derived from the tests carried out to validate the orthodox p-N formalism. In this sense these corrections are remarkable [3].

## 2. Materials and Methods

As already mentioned, the code contains the N orbital equations for a spherical Earth and four ODE p-N systems of orbital equations. Two of these systems are ECI equations for the trackers and targets, say $S$ and $D$, and the other two describe the relative motions of $D$ with respect to $S$ (whose results have to be compared with the N results). The first two p-N systems are orthodox systems, i.e., they are derived from the geodesic hypothesis for the second order p-N approximation of the Earth Schwarzschild field, the third from the standard ECI relative motion of $D$ with respect to $S$, and the last from Synge's equations for the relative motion of $D$ with respect to $S$ referred to INSs on board of $S$ [4].

The second order p-N approximation of the Earth Schwarzschild field for an isolated Earth in curvature coordinates, $(x^\alpha, x^4)$ reads ($\alpha = 1, 2, 3$, and $x^4 \equiv t$, where $t$ is the proper time of a particle at rest at infinity)

$$g_{\alpha\beta} = \delta_{\alpha\beta} + \left(\frac{2m}{r} + \frac{4m^2}{r^2}\right)\frac{x_\alpha x_\beta}{r^2} + O\left(\varepsilon^3\right), \ g_{\alpha4} = O\left(\varepsilon^{5/2}\right), \ g_{44} = -1 + \frac{2m}{r} + O\left(\varepsilon^3\right), \ (1)$$

where $m$ is the mass of the Earth, $r^2 = x^\alpha x_\alpha$ ($m$ and $r$ are in seconds—see below) and $\varepsilon = O(m/r)$.

Thus, for the first two systems we have [5]

$$\frac{d^2 x^\alpha}{ds^2} = \frac{-mx^\alpha}{r^3}\left[\left(2\delta_{\beta\gamma} - \frac{3x^\beta x^\gamma}{r^2} - \frac{2m}{r}\cdot\frac{x^\beta x^\gamma}{r^2}\right)\frac{dx^\beta}{ds}\cdot\frac{dx^\gamma}{ds} + \left(1 - \frac{2m}{r}\right)\left(\frac{dx^4}{ds}\right)^2\right] + O\left(\varepsilon^3\right), \quad (2)$$

while the relations of the respective proper times, $s$, of $S$ and $D$, and the coordinate time, $t$, are given by

$$\frac{d^2 x^4}{ds^2} = \frac{d^2 t}{ds^2} = \frac{-2mx^\gamma}{r^3}\left(1 + \frac{2m}{r}\right)\frac{dx^\gamma}{ds}\cdot\frac{dx^4}{ds} + O\left(\varepsilon^3\right). \quad (3)$$

It holds up to $O\left(\varepsilon^2\right)$ that

$$\frac{ds}{dt} = 1 - \frac{m}{r} - \frac{1}{2}(v)^2, \quad (4)$$

where $x^\alpha$, $r$ and $v$ are ECI magnitudes alternatively referred to $S$ and $D$. Consequently, the closest values to $t$ are realized by TAI, since the Earth is considered isolated here [6]. (Neglecting the $O\left(\varepsilon^2\right)$-terms, the metric (1) can be found in [7] and, in isotropic coordinates, in [6] and [8]. The equations in standard p-N coordinates can be derived from the alternative metric in [9]).

The fourth system has the form

$$\frac{d^2 X^{(\alpha)}}{ds^2} = -X^{(\gamma)}\int_0^1\left(1 - 2u + 3u^2\right)R_{(\alpha4\gamma4)}du + X^{(\mu)}X^{(\nu)}\int_0^1 (1-u)u^2\frac{\partial R_{(\mu4\nu4)}}{\partial x^\alpha} + O\left(\varepsilon^3\right), \quad (5)$$

where $X^{(\alpha)}$ are the p-N coordinates of $D$ with respect to $S$ related to an INS co-moving with $S$. Now we arrive at the following conclusion [10]:

System (5) describes the relative motion of $D$ with respect to $S$ with the highest accuracy. It is the only system that involves line integrals of the Riemann tensor along the straight lines connecting $S$ and $D$ (of course, there are simpler systems but less accurate [11]). Hence, it is only solved when $D$ is in the line of sight (LOS) of $S$, i.e., when $D$ is not eclipsed to $S$ by the Earth, nor shadowed by the Atmosphere. This implies that the matching between the two systems of relative equations is made when $D$ is clearly seen by $S$.

The integration of all ODE systems is carried out using the ode45 MATLAB solver with absolute and relative tolerances $10^{-16}$ and $10^{-13}$, respectively. The relative tolerance

for the numerical evaluation of the integrals involved in (5) is $10^{-16}$. To calculate the optimal shooting instants, we introduce suitable event functions.

To be coherent with the tracking measurements, we adopt for the p-N equations the standard p-N initial conditions that correspond to the N conditions (see e.g., [12]) and all quantities and constants are measured in seconds, as in [4]. In fact, we rescale the original quantities $c = 2.99792458\text{e}10 \text{ cm s}^{-1}$ and $G = 6.67430\text{e} - 8 \text{ g}^{-1} \text{ cm}^3 \text{ s}^{-2}$ to $c = G = 1$ so that the mass and radius of the Earth become $m = M = 1.47936611\text{e} - 11 \text{ s}$ and $R = 2.125\text{e} - 2 \text{ s}$, respectively. Then, for reader convenience, the numerical results, thus derived in seconds, are recalculated into IS units (see figures and Table 1).

Finally, to start the computations for autonomous tracking, we assume that $S$ is maneuvered until the orbital elements of $S$ and $D$ coincide, except for the orbital eccentricity and the semimajor axis. The semimajor axis of $S$ is required to be larger than the one of $D$. According to [13], for $S$ to become autonomous the perigee of the orbit of $S$ is assumed to be aligned with the perigee of the orbit of $D$ and with the ECI center. The eccentricity of $S$ is then adapted in such a way that $S$ and $D$ have the same velocity at perigee (If the orbit of $D$ is circular, then the integration starts when $S$ is at perigee and $S$ and $D$ are aligned with the ECI center.).

The freedom to select the initial tracking distances for these maneuvers, whether autonomous or not, is limited by the following (golden) rule:

(i) any efficient and safe deflection is performed when $D$ approaches $S$ from behind within the same orbital plane; (If the orbit of $D$ is circular, then the integration starts when $S$ is at perigee and $S$ and $D$ are aligned with the ECI center.)

(ii) the smaller the initial distance between $S$ and $D$, the longer $S$ takes to reach the optimal instant to shoot $D$ down.

In addition to this rule, we have that only the laser capabilities of $S$ and factors like the size of $D$, their degree of threat, and their material composition limit the choice of the initial range from $S$ to $D$. Indeed, we assume our initial data (altitude at perigee and eccentricity) and test our equations on it.

From the astrodynamic standpoint, we can say that for $S$ to succeed, it is necessary to shoot at $D$ when the p-N direction from $S$ to $D$ is opposite to the p-N ECI velocity of $D$. In other words, when the transverse velocity of $D$ with respect to $S$, as measured by the INS on board of $S$, is zero. Our goal, therefore, is to determine the shooting instants and pointing directions corresponding to these optimal configurations.

In the next section, we discuss numerical simulations to illustrate our findings.

## 3. Numerical Simulations

In Table 1, we consider six representative scenarios, which involve deflections of middle size LEO debris objects (those between 1 cm and 10 cm). According to the previous section, we do not include considerations on the physical characteristics of $D$ and focus our attention on shooting accuracy. We are in particular interested in showing the influence of the initial difference between the altitudes of $S$ and $D$ on the waiting time for $S$ to successfully reach $D$.

**Table 1.** Orbital elements of $S$ and $D$ and results.

|  | alt($D$) (km) | e($D$) | alt($S$) (km) | e($S$) | t-Span (h) | Time Diff. (s) | Mean mot. of $D$ (m) | arc Diff. (cm) | rg. Diff. (cm) |
|---|---|---|---|---|---|---|---|---|---|
| **Case1** | 200 | 0.01 | 250 | adapt. | 63.4 | 0.0004731 | 3.67 | 2.8 | 3.9 |
| **Case2** | 200 | 0.01 | 350 | adapt. | 21.3 | 0.0001111 | 0.86 | 2.0 | 6.2 |
| **Case3** | 200 | 0.02 | 400 | 0.06 | 14.1 | 0.0001229 | 0.95 | 3.3 | 29.2 |
| **Case4** | 200 | 0.01 | 350 | 0.02 | 30.1 | 0.0012570 | 9.74 | 15.8 | 5.0 |
| **Case5** | 250 | 0.02 | 450 | 0.1 | 9.4 | 0.0003379 | 2.60 | 14.5 | 23.3 |
| **Case6** | 250 | 0.01 | 400 | 0.12 | 7.7 | 0.0005099 | 3.94 | 26.6 | 49.3 |

We assume *S* and *D* to be in coplanar (equatorial) orbits, since this does not imply loss of generality due to the sphericity of the Earth Schwarzschild field. For this reason, we assume that all orbital elements of *S* and *D* are zero, except for those in Table 1. In particular, the first two cases correspond to autonomous trackers *S*, so that, as was said in Section 2, their orbital eccentricities are adapted to the orbits of the respective *D* [13].

In the first column the altitudes of *D* above the Earth take the values 200 km or 250 km, while in the third column the altitude of *S* varies between 250 km and 450 km, so that the semimajor axis of *S* in each case is larger than the semimajor axis of *D.* In accordance to part (i) of the golden rule, this allows *S* to shoot backwards while *D*, moving faster than *S*, is approaching from behind.

The second and fourth columns contain the respective eccentricities of *S* and *D*. In the first two cases the eccentricities of *S* are adapted to the motions of *D* for *S* to autonomously reach *D*.

In column five we report on the minimum timespans required for each *D* to be at the optimal position with respect to *S*. Here, we observe not only that part (ii) of the golden rule is respected, but also that the timespan required to reach the optimal position may be shorter for trackers working autonomously, see cases 2 and 4.

We now point out to the main results in column six of the table, the differences between the p-N and N optimal shooting instants. The resulting time corrections can be used to reach the highest accuracy in deflecting objects *D*. They also enable the validation of the p-N formalism to describe the Earth surrounding space, see end of Section 1.

These results are complemented by the data in columns 7 to 9, which contain the corrections in the location. The mean motions of *D* from the N instants up to the p-N instants can be found in column 7. The corresponding deviations of the directions and corrections to the ranging from *S* to *D* are in columns 8 and 9, respectively. The values show that the resulting trajectories of *D* could be undesirable if *S* shot *D* down at the N instants instead of the p-N instants. Consequently, the p-N instants lead to the most efficient and most safe way of deflecting *D*.

Figures 1–4 illustrate the outputs in Table 1. To keep the presentation concise, we show the results for Case 2, as a representative for the actions made by autonomous trackers, and Case 4 for trackers that are not autonomous.

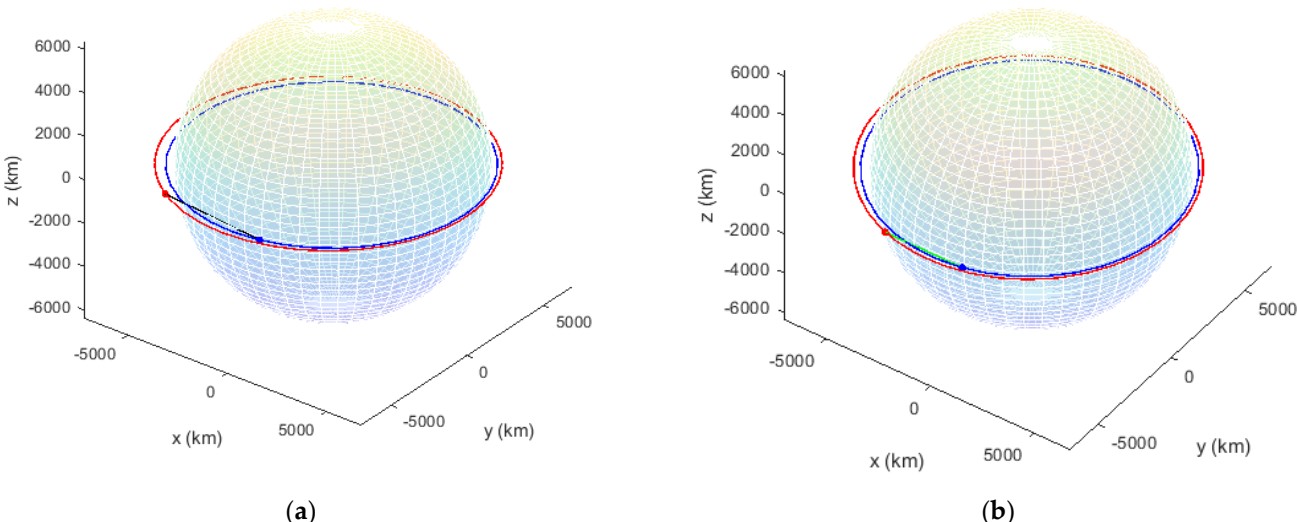

(**a**)                    (**b**)

**Figure 1.** ECI p-N orbits of *S* and *D*: (**a**) Case 2, (**b**) Case 4.

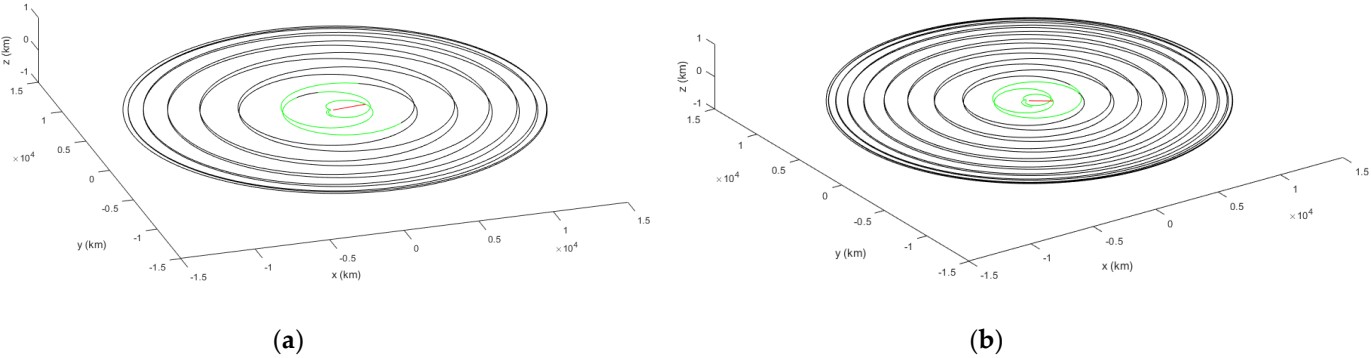

**Figure 2.** Relative p-N orbits of *D* w. r. t. *S* with shooting action: (**a**) Case 2, (**b**) Case 4.

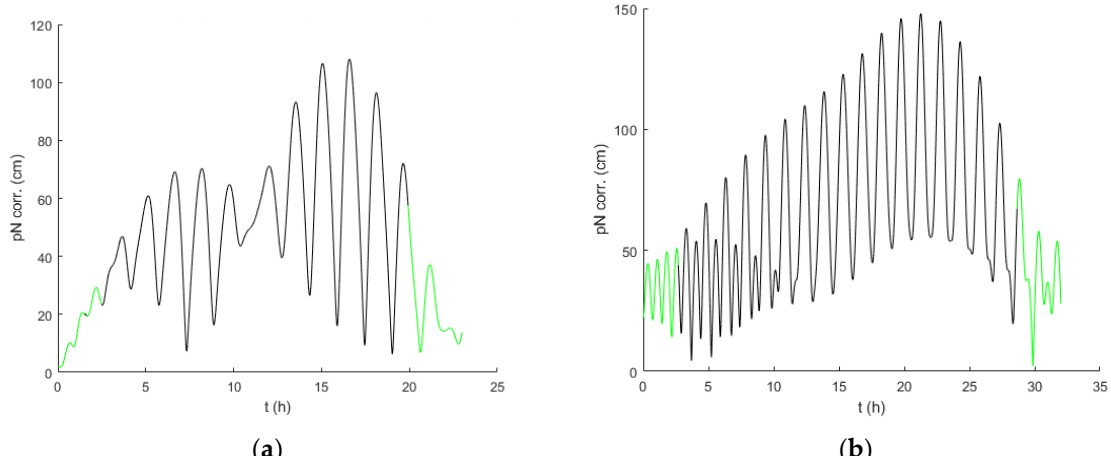

**Figure 3.** p-N corrections to the N locations of *D* w. r. t. *S:* (**a**) Case 2, (**b**) Case 4.

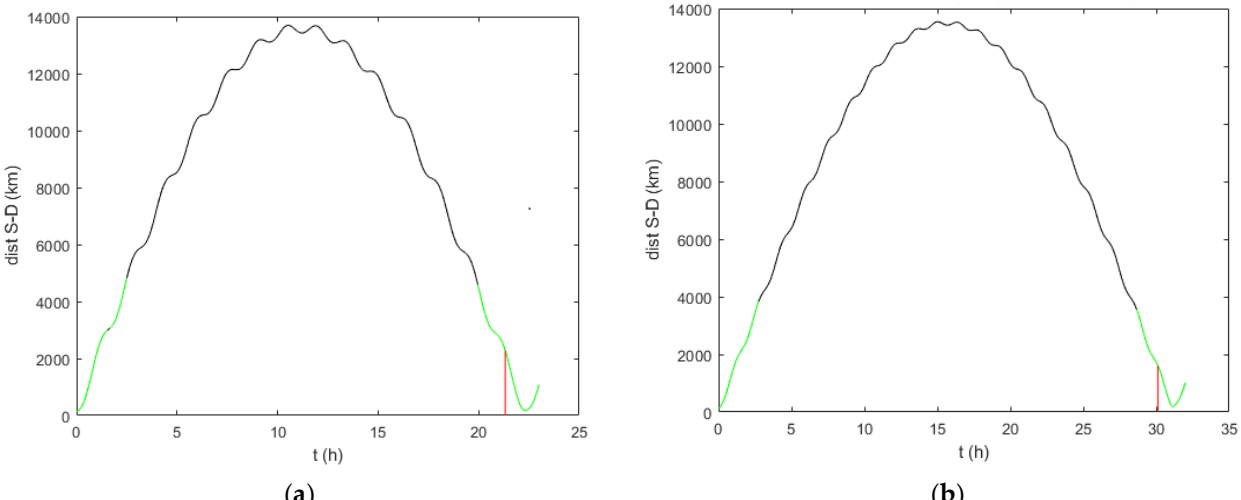

**Figure 4.** N distances from *S* to *D* with shooting actions: (**a**) Case 2, (**b**) Case 4.

Figure 1 (a) Case 2 and (b) Case 4 are snapshots of the p-N evolutions of *S* and *D* during equal timespans (2.7 h). The tracker's trajectories are depicted in red, and the target's in blue. In both cases *D* is ahead of *S*, so that *S* has to wait for *D* to approach from behind. The qualitative difference between these situations is the following: in (a) *D* is already eclipsed to *S* by the Earth (see the black segment *S-D*) while in (b) *D* still can be seen by *S* (see the green segment *S-D*).

Figure 2 (a) Case 2 and (b) Case 4 show the respective relative orbits. We use different colors (green and black) to indicate if there is a clear line of sight between *S* and *D* or if the objects are eclipsed by the Earth. The small red segments *S-D* show the connecting line between *S* and *D* at the p-N shooting instants.

Figure 3 (a) Case 2 and (b) Case 4 depict the p-N corrections to the N locations of *D* with respect to *S*. As before, the colors indicate the position of *D* with respect to *S* to clarify when the fourth system of p-N Equation (5) was solved to compute the p-N relative motion of *D*. Note, that parts of black lines are hidden under the green lines due to the scaling.

In Figure 4 (a) Case 2 and (b) Case 4 the N distances between *S* and *D* during the whole timespans are shown. The colors indicate the times at which *D* is in the LOS of *S*, while the red segments again show the p-N shooting instants. Together with Figure 2 it is clear that the optimal shooting instants occur when *D* is approaching *S* from behind, which can take a long time. This is the price to be paid for shooting in the most safe and efficient way. This could also be observed in the whole animated motion from which the snapshoots in Figure 1 stem.

## 4. Conclusions

From the results in this paper, we can conclude that the computational procedure described here can help increasing the accuracy of shooting down LEO debris targets. The procedure is designed for LEO autonomous trackers to throw the targets into the Atmosphere by means of ablations. In fact, the method manages visibility parameters of the targets, specifying how and when the trackers should carry out the optimal and safe ablation actions. The results in Figure 3 show that the corrections to the N locations (which are currently used) can be larger than the size of many targets. Therefore, after comparing benefit and computational cost, it seems reasonable to include this procedure into those already utilized for tracking.

**Author Contributions:** Conceptualization, J.M.G. and M.L.G.d.P.; methodology, J.M.G., M.L.G.d.P., J.M. and E.B.W.; software, J.M.; formal analysis, J.M.G. and M.L.G.d.P.; writing—original draft preparation, J.M.G. and M.L.G.d.P.; writing—review and editing, J.M. and E.B.W. All authors have read and agreed to the published version of the manuscript.

**Funding:** This research received no external funding.

**Conflicts of Interest:** The authors declare no conflict of interest.

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
