# Peer review of "Computational Modeling and Simulation to Increase Laser Shooting Accuracy of Autonomous LEO Trackers"

_photonics, doi:10.3390/photonics8020055_

Round 1
Reviewer 1 Report
This paper deals with the post-Newtonian effect correction for laser shooting in LEO.
I have some questions about the concepts:
1) What is the difference between "autonomous tracking" and "non-autonomous tracking"?
2) Does S should have perfect ephemeris data for S and D to apply this method? If so, how would you get that?
3) How would you get the initial tracking distance? Did you just assume that S can be located at its desired position?
Also, here are my comments for the paper organization:
1) More clear description of the problem statement is needed. It is hard to understand the assumptions about S and D and the concept of operations.
2) Please give us a structure of your code (maybe pseudo-code?) for readers' convenience.
3) The p-N equations were presented in Ref.[5]. Then, what is the contribution of this paper? Is it assessing the orbit prediction errors?
Reviewer 2 Report
The paper deals with a subject of growing interest as the problems posed by debris in LEO are started to be recognized as relevant by space agencies. Laser ablation of small scale debris is an option which deserves attention and the shooting accuracy of the trackers is a complex scientific/technological issue. The Authors contribute to the foundational knowledge of the process with an accurate model that accounts for all the parameters that enter into the game. The research methodology is grounded on a solid theoretical background and the essential results are presented in order to not make heavy the reading. The numerical results are clearly displayed. The article may trigger a wider interest in the scientific community on such a visionary matter and stimulate further investigations.
Reviewer 3 Report
The subject of this paper is very important and actual. In general, this paper is quite interesting but I see some room for improvement. Below you will find a list of my remarks and questions: 1. Figures 1 and 2, no captions are given for z-axes. No unit is stated for numbers on z-axes. 2. Figure 1, tracker’s trajectories and target’s trajectories are very close to each other and it is hard to distinguish them. Maybe enlarging of (a) and (b) parts of this figure would improve this situation. 3. Figure 4, please write what is the meaning of red vertical lines in both parts (a) and (b) of this Figure. 4. ‘Conclusions’ section, line 185, you write: “The results show that the corrections to the N locations (which are currently used) can be larger than the size of many targets.” Where in the paper text, figures or tables you may find such comparison? You presented some results but in my opinion they are poorly described and more detailed discussion on results is necessary. 5. ‘Conclusions’ section, line 186, you write: “Therefore, after comparing benefits and cost, it seems reasonable to include this procedure into those already utilized for tracking.” Not a single word is written about costs. How big reduction of costs is possible to be achieved by implementation of presented procedure?Author Response
Please see the attachment

Round 2
Reviewer 1 Report
Accepted
Reviewer 3 Report
I have no more negative comments.